# Assessment of neonatal thermal cares: Practices and beliefs among rural women in West Guji Zone, South Ethiopia: A cross-sectional study

**Wako Golicha Wako**[1]*, **Belda Negesa Beyene**[2], **Zelalem Jabessa Wayessa**[2], **Aneteneh Fikrie**[1], **Elias Amaje**[1]

1 Bule Hora University, Institute of Health, School of Public Health, Bule Hora, Ethiopia, 2 Department of Midwifery, Bule Hora University, Institute of Health, Bule Hora, Ethiopia

* bwakot730@gmail.com

## Abstract

A newborn has a limited capacity to maintain temperature when exposed to cold environment. Neonatal hypothermia, a common neonatal problem, carries high case fatality rate particularly if concurrently occurs with other neonatal problems. This study assessed neonatal thermal care practices and beliefs among rural women in west Guji Zone, south Ethiopia. A community based cross sectional quantitative study combined with qualitative study was undertaken in rural areas of west Guji Zone, Ethiopia. Randomly selected 388 rural mothers of infants less than 6 months old were participated in the quantitative study. Three focus group discussions were conducted among mothers of infants less than 6 months old. Quantitative data were collected by using structured and pretested Afaan Oromo version questionnaire adapted from relevant literatures. Qualitative data were collected by focus group discussion guide. The quantitative data were cleaned, coded and analyzed by SPSS version 20. Qualitative data were transcribed, translated, coded, and analyzed by thematic analysis approach. In general rural women believe that thermal protection of newborn is important. The findings show that approximately 75% and 85% of newborns were dried and wrapped respectively after delivery. However drying and wrapping of newborn are usually done after the first newborn's bath. Just over 84% of newborns were bathed within the first 6 hours of delivery and majority of them were bathed with warm water. About 69.1% and 57.7% of women put head cover to their newborns immediately after birth, and initiated breast feeding within one hour of delivery respectively. Skin to skin care of newborn is non-existent in the study area and perceived as an odd, frightening and potentially dangerous practice. Studied women practice some of the recommended neonatal thermal cares and believe in their importance in keeping newborn warm. However, practice and beliefs about delayed first bath is against standard recommendation, whereas skin to skin care is non-existent and perceived as an odd practice. Interventions to familiarize skin to skin care and reduce misconceptions surrounding it should be introduced into the study area to improve thermal cares of high risk newborns.

**Data Availability Statement:** All relevant data are included into the manuscript including tables.

**Funding:** This study was financially funded by Bule Hora University for WGW with grant number PRD/

124/11. No additional financial or material assistance was obtained for this study from other sources. The granted finance was only utilized for payment for data collectors. All authors are employees of Bule Hora University and they were paid monthly salary as employees. But they did not receive payment from the University as part of the research work. The funder had no role in study design, data collection and analysis, decision to publish or preparation of the manuscript.

**Competing interests:** The authors have declared that no competing interests exist.

## Introduction

Newborns have a limited capacity to adapt to a range of environmental temperatures. They have a limited ability to dissipate heat effectively in warm environments. More critically, they are unable to maintain temperature when exposed to cold environments. Therefore it is important to keep them in optimal thermal environment [1]. Neonatal hypothermia occurs when newborn's body temperature drops below lower limit of normal range of 36.5–37.5˚c. A newborn with temperature range of 36–36.4˚c may be under cold stress; 32–35.9˚c has moderate hypothermia and below 32˚c is considered to be in severe hypothermia [2].

Neonatal hypothermia is a common problem of newborns, and in some settings, as high as half of all newborns become hypothermic immediately after birth [3,4]. Mild to moderate hypothermia is nearly universal, with substantially higher risk in cold seasons. However, its incidence in hot seasons is also high; thus, year-round thermal care protection is required [5]. Neonatal hypothermia is one of major public health problems in Sub-Saharan Africa [6]. In Africa, prevalence of newborn hypothermia is high, ranging from 44% to 85% [7]. Though all newborns are at risk of losing heat after birth; preterm and/or small for gestational age newborns are particularly vulnerable to hypothermia [7]. In Sub-Saharan Africa, 12.3% of newborns are born preterm and 25.5% of newborns are small for gestational age (SGA) [8].

Global case fatality rate of neonatal hypothermia ranges from 8.5 to 52%, and the fatality rate is higher when hypothermia concurrently occurs with other neonatal problems [7]. Hypothermia rarely causes direct death, however it significantly contributes to neonatal mortality if concurrently occurs with severe infections, asphyxia and preterm birth [7]. It is associated with nearly fivefold increased risk of deaths within the first 7 days of life [9]. As significant proportion of child mortality happens during neonatal period, more rapid reduction of neonatal death is required to improve child survival [10,11]. Appropriate thermal protection could be one of the important interventions to achieve faster reduction of neonatal deaths. Three thermal care practices (delayed bathing, head covering and skin to skin care) could avert up to 20% of neonatal deaths caused by preterm birth complications and 10% of newborn deaths due to infections among full term or moderately preterm neonates [12]. In sub-Saharan Africa, importance of neonatal thermal care is well recognized [13]. However neonatal thermal care practice is suboptimal in the region [13]. Harmful cultural norms and traditional practices that affect neonatal thermal care practices at home exist in different contexts of Africa [14]. These include social pressure to perform early bathing; waiting placental delivery to institute thermal protection measures and negative perception about vernix which impose early bathing of newborns [15]. The lack of thermal protection is still an unrecognized major challenge for neonatal survival in developing countries [7].

Ethiopian government has recognized thermal protection of newborn as one of important child survival interventions [16]. Thermal protection of newborns has been incorporated into essential newborn care package in both community and health facility settings [16]. No previous study has assessed neonatal thermal care practices in rural Ethiopia. Therefore this study assessed practices about neonatal thermal cares (early drying and wrapping, head covering, skin to skin care, newborn bath time and early breast feeding initiation) and related beliefs among rural women in West Guji Zone, South Ethiopia.

## Methods and materials

### Study setting and period

The study was conducted in West Guji zone, South Ethiopia from October 2018 to April 2019. West Guji zone's administrative town, Bule Hora is located 475 km south to Ethiopia's capital,

Addis Ababa; at latitude and longitude of 5˚35'N 38˚15'E and 5.583˚N 38.250˚E respectively. The town is located at an elevation of 1716 meters above sea level and has an average annual rainfall of 648mm. According to 2007 Ethiopian census, the town had a total population of 27,820. With estimated annual growth rate of 2.6%, the population of the town is expected to be approximately 285,433 at the end of 2017.

## Study design

A community based quantitative cross sectional study combined with qualitative study was done. We conducted quantitative cross-sectional study to assess neonatal thermal care practices. Focus groups discussions were conducted to supplement quantitative study findings with regard to neonatal thermal care practices, and also to assess rural women's beliefs about neonatal thermal care practices. Both quantitative and qualitative studies were done at the same time, and they were collected simultaneously.

## Study population

Participants of quantitative study were rural women who have an infant younger than six months. Focus group discussions were also conducted among rural women who have an infant younger than six months.

## Exclusion criteria

We excluded women who gave birth to their youngest infant by cesarean section and/or have undergone major surgical interventions immediately after delivery from the quantitative study. We excluded them because effects of anesthesia associated with such surgical interventions may compromise their ability to recall thermal cares provided to their newborns immediately after delivery.

## Study variables

We assessed neonatal thermal cares focusing on practices and beliefs of rural women as related to immediate drying and wrapping, head covering, skin to skin care, time of newborn's bath, and time of breast feeding initiation after delivery.

## Sample size calculation

Sample sizes for quantitative study were calculated by single proportion formula with the following common assumptions:

- An anticipated sample proportion (p) of 50%, as there is no previous similar study done in rural Ethiopia. We assumed prevalence of early wrapping and drying; delayed newborn baths at least by 6 hours; head covering during neonatal period, skin to skin care, breastfeeding initiation within 1 hour of delivery to be 50%;

- Level of confidence of 95% and

- Margin of error (w) of 5%.

Then the sample size was calculated using the formula:

$$n = Z^2\, p(1-p)/w^2;$$

Where n- is minimum required sample size; Z- is z-score corresponding to two sided 95% confidence level; p- is anticipated sample proportion and w- is margin of error. After inputting the

numbers into the formula the minimum required sample size became 384 women. Then adding 5% non-response rate, the final require sample size became 384+384(0.05) = 423 women. This sample size was used for quantitative study. Final data were successfully collected from 388 women and the results of quantitative study is based the data from these women.

Total of 3 focus group discussions (FGDs) were conducted; one focus group discussion from each of three climatic areas of the zone (i.e. Kola, Woinedega and Dega climate areas). We did not consider additional FGDs because we have reached saturation of ideas. All FGDs had 8–10 participants.

## Sampling procedures

Sample size of the quantitative study was distributed to three climatic areas (i.e. Dega, Woinedega and Kolla) within the study area proportional to the approximate population of each climatic area. From each climatic area, one kebele (i.e. the smallest administrative unit) was randomly selected. Lists of all eligible women within selected kebeles were received from health posts within the kebeles. Then required sample sizes from each kebele were randomly selected from lists of eligible women within the kebele. Participants of FGDs were purposively selected to ensure variations in age, socio-economic status and level of education.

## Data collection instruments and procedures

Quantitative data were collected by using structured and pretested Afaan Oromo (i.e. local language of the study area) questionnaire (S1 and S3 Files) adapted from relevant literatures. Data collectors were high school graduate local females who fluently speak Afaan Oromo. Quantitative data were collected at home of study participants. Focus group discussions were moderated by principal investigator. Focus group discussion guide (S2 File), which has open ended questions prepared in local language (i.e. Afaan Oromo) was used to facilitate focus group discussions. Focus group discussion guide questions were developed after reviewing literatures on the area of interest. Focus group discussions were conducted in Afaan Oromo, the language native to all FGD participants, facilitator and moderator. They were conducted in community setting, where only FGD participants, facilitator and moderated were presented. All FGDs took between 90–120 minutes. The discussion sessions were tape recorded and note taking was done. The FGD participants were not compensated for their participation. Transcription and translation of all FGDs were done immediately after the discussion sessions.

## Quality assurance

Training was provided for data collectors and supervisors. Investigators supervised whole quantitative data collection processes. Survey questionnaire was pretested among population with similar socio-demographic characteristics with the current study population outside the sites where the current study was conducted. Adapted English questionnaire was translated into Afaan Oromo language and retranslated back into English by two independent language experts who were judged to be fluent in both languages. All necessary inputs from the pretest were incorporated into final questionnaire. FGD data were transcribed, and then translated into English language by principal investigator. Then the translated notes were checked against audio records and field notes by one member of research team. Any error found in the transcript and translated note was corrected. Two members of research team extensively read translated content and coded data together.

## Data processing and analysis

Quantitative data were entered into Epi-info 7 and exported to Statistical Package for Social Sciences (SPSS) version 20 for further processing. The data were cleaned, coded and analyzed by the SPSS. Frequency of different thermal care practices was generated. Qualitative data were transcribed, translated and entered into open code software. After reading qualitative notes, two investigators coded the data together by emerging themes. The qualitative data were synthesized and analyzed by thematic analysis approach. Major findings from focus group discussions were summarized into five themes. These themes are: importance and timing of drying and wrapping newborn; early bathing of newborn; importance and timing of head covering; useless of breast feeding to keep baby warm; and skin to skin care of newborn as an odd and frightening practice.

## Ethics approval and consent to participate

Ethical clearance was obtained from institutional review board of Bule Hora University (Approval number: PRD/124/2011). All methods were performed in accordance with a proposal approved by Bule Hora University. Informed written consent was obtained from all adult subjects. Informed verbal consent was obtained from parents or legal guardians for subjects less than 18 years old. We prepared the written consent form attached with survey questionnaire which was received from study participants. We did not prepare separate written consent form for parents or legal guardians as we believed only small proportion of study participants were minors. However the informed verbal consent from parents or legal guardians was implemented after approved by institutional review board of Bule Hora University. The verbal consent was documented on separated verbal consent form prepared for this purpose. Confidentiality of the information was secured by omitting personal modifiers on the questionnaire.

## Results

### Socio-demographic characteristics of respondents

Out of the total 423 women planned for the study, 388 women were successfully interviewed with the response rate of 91.7%. Thirty five (35) women were not found within their home for the interview after two home visits. All of the 388 interviewed women were included into the final analysis. The mean (±SD) age of the respondents was 25.4(±5) years. At the time of data collection, the mean (±SD) age of the infants for whom thermal cares were assessed was 3.8 (±1.6) months. Majority of participated women, [283(72.9%)] were literate (i.e. can read and write), and the rest 105 (27.1%) of them were illiterate (i.e. cannot read and write). Protestant Christian was the commonest religion; followed by Islam, Orthodox Christian and Waqefata (i.e. traditional belief); each accounted 251(64.7%), 40(10.3%), 36(9.3%) and 36(9.3%) respectively. Regarding the marital status of women, 325(83.8%) were married, 25(6.4%) were widowed, 17(4.4%) were separated, and 16(4.1%) were divorced. Table 1 presents socio-demographic characteristics of the respondents.

### Past obstetrics history of studied women

At the time of survey, majority of respondents, 311(80.1%) gave birth more than once. Significant proportion of women, 250(64.4%) delivered their youngest infant at home and the remaining 138(35.6%) of women delivered in health facilities. Among these deliveries, only 120(30.9%) of them were attended by health workers of different level of qualification.

**Table 1. Socio-demographic characteristics of the respondents, West Guji Zone, Oromia; Southern Ethiopia; April 2019.**

| Characteristics of respondents | Frequency | Percentage (%) |
|---|---|---|
| **Age of women in completed years (n = 388)** | | |
| <18 | 9 | 2.3 |
| 18–34 | 345 | 88.9 |
| 35–49 | 34 | 8.8 |
| **Women's educational level (n = 388)** | | |
| No education(grade 0) | 126 | 32.5 |
| Primary education(grade 1–8) | 221 | 56.9 |
| Secondary education and above (grade 9 and above) | 41 | 10.6 |
| **Women's occupation(n = 388)** | | |
| Farmer | 165 | 42.5 |
| House wife | 75 | 19.4 |
| Merchant | 105 | 27.1 |
| Daily laborer | 23 | 5.9 |
| Employed | 16 | 4.1 |
| others | 4 | 1.0 |
| **Women's Religion (n = 388)** | | |
| Waqefata | 36 | 9.3 |
| Protestant Christian | 251 | 64.7 |
| Orthodox Christian | 36 | 9.3 |
| Islam | 40 | 10.3 |
| Catholic Christian | 23 | 5.9 |
| Others | 2 | 0.5 |
| **Sex of the youngest infant** | | |
| Male | 213 | 54.9 |
| Female | 175 | 45.1 |

However, majority of these women [298(76.8%)] attended antenatal care during pregnancy of their youngest infant. Table 2 presents past obstetric history of the study participants.

## Early bathing, drying and wrapping of newborns

Despite majority of women preparing clothes for drying (90.2%) and wrapping(86.6%) of new-borns before delivery, only 32(8.2%) of newborns were dried before delivery of the placenta, and 133(29.1%) of newborns were wrapped within 5 minutes of delivery. A significant proportion of women, 266(68.5%) performed drying of their babies after placental delivery respectively. A qualitative finding shows that drying and wrapping are usually done after the first newborn bath in community settings. However, if delivery is conducted in a health facility, or if placental delivery is delayed, newborns are immediately dried and wrapped without having the first bath. An 18 years old mother of five months old infant said "*immediately after delivery before drying, newborns are bathed and then wrapped with clean clothes. But if delivery is at a health facility, newborns are not bathed. They are wrapped with clothes and no bath is done until mothers return their home from the health facility*". Here are other quotes from focus group discussants "*Some women perform drying and wrapping. But this practice is at its early state of expansion among women who have exposure to health facilities. The common practice is to bathe the baby immediately after delivery with warm water, and then wrap it with warm clothes.*" (18 years old and mother of 2 months old infant)...."*nobody pays attention to baby's*

**Table 2.  Past obstetrics history of rural women, West Guji Zone, Oromia; southern Ethiopia, April 2019.**

| Past obstetrics history | Frequency | Percentage (%) |
|---|---|---|
| **Number of all previous births(n = 388)** | | |
| 1 | 77 | 19.8 |
| 2–4 | 187 | 48.2 |
| >= 5 | 124 | 31.9 |
| **Did you seek ANC during pregnancy of the youngest infant(n = 388)** | | |
| No | 90 | 23.2 |
| Yes | 298 | 76.8 |
| **Place of the birth of the youngest infant (n = 388)** | | |
| Home | 250 | 64.4 |
| Health post | 22 | 5.7 |
| Health center | 28 | 7.2 |
| Hospital | 84 | 21.6 |
| Others | 4 | 1.0 |
| **Attendants of the last birth(n = 388)** | | |
| Traditional birth attendant | 38 | 9.8 |
| Relative | 162 | 41.7 |
| HEW* | 40 | 10.3 |
| Health professional | 80 | 20.6 |
| Nobody | 65 | 16.7 |
| Others | 3 | 0.8 |
| **Gestational age of the youngest infant (n = 388)** | | |
| <9 months | 6 | 1.5 |
| >= 9 months | 382 | 98.5 |

*: HEW- Health Extension Worker.

bath before placental delivery. Babies are simply wrapped with clothes without having a bath until placental delivery."(29 years old and mother of 3 months old infant). These findings are in congruent with quantitative findings, which reported waiting for newborns to get bathed as one of the reasons for delayed drying and wrapping (Table 3).

Although FGD participants mentioned cold prevention as the reason for performing drying and wrapping, they believe that drying with dry clothes damages the delicate skin of newborns. Instead of immediate drying, immediate bathing is done to remove birth secretions. A focus group discussant said "*because newborn's skin is very fragile, it should not be touched with clothes for drying. That is why bath is done to remove birth secretions.*"(36 years old and mother of 4 months old infant).

The great majority, 328(84.5%) of women have bathed their baby in less than 6 hours of delivery, and 306(78.9%) of women used warm water for the first bath. Forty two (70.0%) of women who delayed the first newborn bath at least by 6 hours after delivery did so because they were advised by health workers to do so, whereas only 8(11.7%) of them delayed the first bath because they believe it keeps newborns warm. Table 4 presents newborns' first bath time among rural women in west Guji Zone.

The first newborn bath is usually done after placental delivery. The focus group discussants said "*newborns are not bathed before placental delivery.*"(25 years old and mother of a 3 months old infant). . ."*newborns are bathed after placental delivery. Before placental delivery nobody bathes baby.*" (19 years old and mother of a 5 months old infant).

**Table 3. Newborn's immediate drying and wrapping practices among rural women in West Guji Zone, Oromia; southern Ethiopia, April 2019.**

| Newborn drying and wrapping practices | Frequency | Percentage (%) |
|---|---|---|
| **First drying time (n = 388)** | | |
| Immediately before placental delivery | 32 | 8.2 |
| Immediately after placental delivery | 131 | 33.7 |
| A long time after placental delivery | 137 | 35.3 |
| Cannot remember | 88 | 22.7 |
| **First drying time in minute(n = 388)** | | |
| In less than 5 minutes | 92 | 23.7 |
| Between 5–15 minutes | 125 | 32.2 |
| Between 16–30 minutes | 51 | 13.1 |
| After 30 minutes | 57 | 14.7 |
| Could not remember | 63 | 16.2 |
| **Reason for delayed drying (n = 233)** | | |
| Attendant was focusing on mother until placenta delivery | 80 | 34.3 |
| Waited until umbilical cord is cut | 17 | 7.3 |
| Waited until placental delivery | 74 | 31.8 |
| Drying baby is taboo until placental burial | 27 | 11.6 |
| Nobody was available | 21 | 9.0 |
| Waited until baby get bath | 14 | 6.00 |
| **First wrapping time in minute(n = 388)** | | |
| In less than 5 minutes | 113 | 29.1 |
| Between 5–15 minutes | 68 | 17.5 |
| Between 16–30 minutes | 38 | 9.8 |
| After 30 minutes | 117 | 30.1 |
| Could not remember | 52 | 13.4 |
| **Reason for delayed wrapping(n = 223)** | | |
| Attendant was focusing on mother until placenta delivery | 45 | 20.2 |
| Waited until umbilical cord is cut | 28 | 12.5 |
| Waited until placental delivery | 43 | 19.3 |
| Wrapping baby is taboo until placental burial | 6 | 2.7 |
| Nobody was available | 15 | 6.7 |
| Waited until baby get bath | 86 | 38.6 |

It is believed that an early newborn bath before drying and wrapping followed by wrapping with warm and clean clothes prevents cold. Focus group discussants shared "*early bathing is better than drying and wrapping in preventing cold." (*20 years old and mother of 5 months old infant). . ."*Early bathing and then wrapping with clean clothes prevents cold.*"(18 years old and mother of 5 months old infant).

The main reason for performing early bath of newborns is to remove birth fluids from newborns as these fluids have a bad odor and hated by people. It is also believed that if not removed, birth secretions expose newborns to cold. "*Women hate the smell of birth fluids and say that these fluids attract cold. That is why immediate bathing with warm water is common*", said 22 years old and mother of 6 months infant. Similarly, 18 years old and mother of 3 months infant shared "*I am not convinced with the concept of delay bathing. For example, do you feel good if a baby stays without getting cleaned from birth's blood for a day*?" Therefore, to avoid bad odor and cold, immediate bathing is done. FGD participants mentioned two main reasons for immediate newborn bath, which are directly or indirectly related to cold

**Table 4. Newborns' first bath time among rural women in west Guji Zone., Oromia; southern Ethiopia, April 2019.**

| The first newborn bath practices | Frequency | Percentage (%) |
|---|---|---|
| **The first bath time after delivery (n = 388)** | | |
| In less than 6 hours | 328 | 84.5 |
| after 6 hours | 60 | 15.5 |
| **Type of water used for the first bath (= 388)** | | |
| Warm water | 306 | 78.9 |
| Cold water | 61 | 15.7 |
| others | 21 | 5.4 |
| **Reason for delayed newborn bath(n = 60)** | | |
| To keep baby warm | 8 | 13.3 |
| Health professional advised me to do so | 42 | 70.0 |
| Others | 10 | 16.7 |
| **Reason for early newborn bathing (n = 328)*** | | |
| To remove different secretions or blood or dirty from body | 270 | 82.3 |
| To remove visible vernix | 62 | 18.9 |
| To improve health and strength of newborn | 18 | 5.5 |
| To make baby refreshed and comfortable | 26 | 7.9 |
| Encouraging sleep and improving health | 15 | 4.6 |
| To prevent different smells | 33 | 10.1 |
| Others | 4 | 1.2 |

* The percentage may not add up to 100 as responses are not mutual exclusive.

prevention. The first reason is to remove malodorous birth secretions so that drying and wrapping can be done to prevent cold. Similar to this, 82.3% of survey participants mentioned that they bathed newborns early to remove different secretions or blood or dirty secretion from the body. The second reason is the belief that if not removed, birth secretions attract cold. Therefore women perform early bath to prevent cold.

Early bathing of newborns as soon as possible after placental delivery is deep-rooted practice. Women strongly believe that newborns should get a bath as soon as possible after placental delivery. Focus discussants said "*if placenta is safely delivered, the newborns should be bathed immediately and kept away from cold.*"(25 years old and mother of 5 months old infant)... "*If placenta is immediately delivered, then the baby will be bathed immediately. Then, there will be no problem of cold.*"(22 years old and mother of 5 months old infant).

## Head covering and immediate breastfeeding of newborns

Two hundred sixty eight (69.1%) women put head covering to their baby immediately after delivery and almost all of them [267(68.8%)] maintained the practice during neonatal period. Qualitative assessment shows that head covering is done to prevent cold as well as sun exposure. Two hundred sixty six (68.6%) women had prepared head covering before delivery. All women believe in the importance of head covering to prevent cold from newborns. Moreover, immediate covering of head of newborns as soon as possible is believed to be an important practice. Focus group discussants shared "*Yes it* [head covering] *is helpful. It prevents cold from injuring baby.*" (33 years old and mother of 5 months old infant)... "*Head covering should be done immediately after delivery.*" (28 years old and mother of 5 months old infant).

**Table 5. Rural women's practice of breast feeding during neonatal period, West Guji Zone, Oromia; southern Ethiopia, April 2019.**

| breast feeding practices | Frequency | Percentage (%) |
|---|---|---|
| **Time to initiate breast feeding after delivery (n = 364)** | | |
| Within one hour | 210 | 57.7 |
| after one hour | 154 | 42.3 |
| **What did you do with colostrum? (= 364)** | | |
| Provided to baby | 136 | 37.4 |
| Discarded | 226 | 62.1 |
| others | 2 | 0.6 |
| **Did you provide extra foods/fluids during neonatal period (n = 388)** | | |
| Yes | 137 | 35.3 |
| No | 227 | 58.5 |
| Could not remember | 24 | 6.2 |
| **Reasons for delayed initiation of breast feeding within 1 hour of delivery (n = 154)** | | |
| Colostrum is dirty/not good for baby | 28 | 18.2 |
| Lack of sufficient breast milk | 10 | 6.5 |
| Baby need sleep/rest after delivery | 55 | 35.7 |
| Baby did not show sign of hunger | 10 | 6.5 |
| Baby has to be bathed | 13 | 8.4 |
| Mother has to rest | 18 | 11.7 |
| Mother needed bathing | 10 | 6.5 |
| Others | 10 | 6.5 |

Three hundred sixty four (93.8%) women breastfed their baby, and 210(57.7%) of them initiated breast feeding within one hour of delivery. Exclusive breast feeding during neonatal period was practiced by 227(62.4%) women. The FGD participants did not mention any thermal protection benefits associated with breastfeeding including early initiation and exclusive breastfeeding. The reasons for early initiation of breastfeeding that FGD participants mentioned are that because they see it from their seniors, and they fear that newborns may not adapt to breastfeeding if breastfeeding is delayed. Table 5 presents rural women's breastfeeding practice after delivery in west Guji zone.

Though initiation of breastfeeding is almost universal, and women claim that they initiate breast feeding immediately after placental delivery and after the first bath, they neither mention nor believed in thermal care importance of either early or frequent breast feeding. Focus group discussants quoted "*I do not think that early initiation of breast feeding helps in keeping baby warm except that delayed initiation of breastfeeding results in baby's failure to adapt with breastfeeding.*"(20 years old and mother of 5 months old infant). . . "*Early initiation of breastfeeding does not prevent cold from newborns. Delaying up to 2to 3 hours after delivery without initiating breastfeeding does not have any problem. Just we learned early initiation of breastfeeding from our mothers.*"(25 years old and mother of 5 months old infant).

## Skin to skin care of newborns

Skin to skin care of newborn is non-existent thermal care practice and is strange to women in the study area. Women perceive skin to skin care of newborn as an odd, frightening and potentially dangerous way of handling newborns. Focus group discussants quoted "*such practice [*i.e. skin to skin care*] is not done in this area. Holding baby in such way is frightening. This practice is new to us. We hear/see it today.*"(20 years old and mother of 5 months old infant). . .

"*If skin to skin care is done, babies may get injured especially by inexperienced young mothers.*"(30 years old and mother of 4 months old infant). . ."*Because we are strange to it* [skin to skin care], *it looks somewhat odd to us.*"(20 years old and mother of 3 months old infant). Elder women do not advise younger women to do skin to skin care. Furthermore, women do not believe that skin to skin care helps in keeping newborn warm. However they claimed that they can do skin to skin care of newborns if they are taught to do so.

## Discussion

Studied women collectively agree on importance of thermal protection of newborns. Drying, wrapping, head covering and early initiation of breastfeeding are common neonatal thermal care practices, whereas delayed bathing and skin to skin care are infrequent thermal care practices in the study area. Even though drying and wrapping are common newborn thermal care practices, they are usually performed after the first newborn bath. Local women believe that wrapping, head covering and early bath keep newborn warm, and also they believe better thermal protection is achieved if drying and wrapping are preceded by bath. They do not believe early initiation of breastfeeding and skin to skin care help in keeping newborns warm.

In this study over 75% and 85% of women reported drying and wrapping newborns after delivery respectively. Approximately 23% and 29% newborns were dried and wrapped respectively within 5 minutes of delivery. These findings are somehow similar to a study conducted in Tanzania where more than 75% of newborns were dried and wrapped after delivery [17]. In Tanzania, 42% and 27% of newborns were dried and wrapped within five minutes of delivery respectively [17]. In south Asia, a study reported better drying and wrapping practices, where 81% of newborns in India; 66% of newborns in Bangladesh and 83% of newborns in Nepal received immediate drying and wrapping [18]. In current study area, drying and wrapping are usually done after first newborn bath except in the circumstance of health facility delivery, where limited access to water and/or health workers' advice might have delayed early newborn bath. WHO recommends immediate drying and wrapping of newborns and delaying the first newborn bath by at least 6 hours after delivery [19] as immediate newborn bath caries risk of cold injuries [20]. Though women accept general importance of wrapping in preventing cold, they believe better thermal protection will be achieved if drying and wrapping are preceded by bath.

Early newborn bathing contributes to significant heat loss and increases risk of hypothermia [21,22]. Delaying the first newborn bath by at least 6 hours helps in keeping newborns warm and consequently reduces risks of morbidity and mortality associated with neonatal hypothermia [7]. In this study, only 15.5% of women delayed first newborn bath by at least 6 hours. Previous study also reported that 70% newborns in Ethiopia were bathed within the first 6 hours of delivery [13]. Study conducted in Tanzania reported lower figure, where 59% of newborns were bathed within the first 6 hours of delivery [17]. In Zambia, women participated in a qualitative study stated that newborns are usually bathed immediately after delivery [23]. Similar to previous studies from Sub-Saharan Africa [13,23], main reasons for early newborn bath are to remove different secretions/blood/dirty from newborn's body, to avoid bad smell and remove visible vernix. Studied women do not believe that delaying newborn bath helps in keeping newborn warm; instead they believe early bath helps in keeping newborn warm. This might be due to the fact that nearly 80% of women used warm water for bathing newborns.

Head covering reduces heat loss due to large surface area of newborn's head [2]. Lack of head covering is significantly associated with neonatal hypothermia [24]. In current study, 69.1% of women covered their newborns' head immediately after delivery. Almost all of these

women (68.8%) maintained the practice during neonatal period, and head covering is believed to be an important means of preventing cold injuries.

Delayed initiation of breastfeeding is one of the risk factors for neonatal hypothermia [21,25]. More than half of women (57.7%) participated in the current study initiated breast feeding within one hour of delivery and 37.4% of women fed colostrum to their baby. Similar findings were reported from a systematic review conducted in sub-Saharan Africa (Ethiopia included) [13]. However, in Tanzania, only 18% of women initiated breastfeeding within one hour of delivery [17]. Studied women do not believe that early initiation of breastfeeding does help in keeping newborns warm. They simply initiated breastfeeding early because that was what they had learnt from their grandmothers/elder women.

Skin to skin care is non-existent in the study area and perceived as an odd and strange thermal care practice. Studies conducted in Sub Saharan Africa reported similar findings [13,23]. In Zambia skin to skin care is not practiced [26]. Similarly, in Tanzania skin to skin care is new to local women and rarely practiced [17,23]. In Ghana 10% of newborns received skin to skin care [27]. However, relatively higher prevalence of skin to skin care of newborns was reported from south Asian countries. For instance in Bangladesh and Nepal about 25% and 38% of women reported engaging in skin to skin care of their newborns respectively [18]. Studied women do not believe that skin to skin care has importance in thermal care of newborns. Furthermore, they also perceive the practice as a potentially dangerous way of handling newborns especially among younger mothers who may not have adequate experience of caring delicate body of newborns. The reported reasons for lack of skin to skin care of newborns in Africa were mainly hypothetical as the practice is rare in Africa [13]. In a previous study, women mentioned potential injuries to newborn's delicate bone, cord and chest problems as hypothetical barriers to skin to skin care of newborns [23]. Exhaustion of mother after delivery; fear of disease from mother to newborn; mother feeling pain/experiencing problems after delivery and few opportunities due to competing activities were also reported as barriers to skin to skin care [13]. Current beliefs and perceptions surrounding skin to skin care might impede future implementation of the practice in the study area.

In the study area, interventions targeting the first newborn bath and skin to skin care of newborns should be incorporated into essential newborn care package to improve thermal care practices. Local women should be educated about thermal care importance of delaying the first newborn bath by at least six hours after delivery. They should also educated and provided with necessary supports to engage in skin to skin care of newborns as an important newborn thermal care especially for preterm and small for gestational age neonates. Misconceptions surrounding skin to skin care of newborns should be targeted by relevant public health interventions.

## Limitations and strengths of the study, and further research needs

This study assessed neonatal thermal care practices of women who gave birth within the last 6 months preceding the survey date. Women were questioned to remember and report actual practices they had done to provide thermal protection to their newborns within 6 months preceding the survey. Therefore recall bias might have compromised accurate reporting of actual thermal care surrounding immediate postpartum period. We could not elicit what had been done to newborns for short period between delivery and the first bath, and thermal practices within that particular period is not known. Therefore future research should explore thermal care practices during short transition period between birth and the first bath in the study area. As qualitative study participants were selected purposively, its findings cannot be generalized to all women in the study area. However, incorporation of the qualitative assessment into the

study has enhanced better understanding of beliefs, perceptions and reasons behind neonatal thermal care practices.

## Conclusion

Studied women practice some of recommended neonatal thermal cares and believe that they are important in keeping newborns warm. However practice and believes about delayed first newborn bath is against standard recommendation, whereas skin to skin care is non-existent and perceived an odd practice. Immediate drying and wrapping are done by significant proportion of women; however their timing is not correct as they are usually performed after the first newborn bath. Though immediate initiation of breastfeeding is common it is not considered by women as an important means of thermal protection.

## Supporting information

**S1 File. English version questionnaire (quantitative tool).**
(DOCX)

**S2 File. Focus group discussion guide (for mothers of infants less than 6 months old).**
(DOCX)

**S3 File. Afaan Oromo version questionnaire (quantitative tool).**
(DOCX)

## Acknowledgments

First of all, we thank Almighty God for giving us strength, patience and endurance during this research undertaking. Next, our thank goes to Bule Hora University for providing financial assistance to complete this research, without which this research could not have been realized. We also acknowledge health officials from district health offices within West Guji Zone and health workers, especially health extension workers who cooperated and provided assistance during field works. Their contributions were very immense. We are also grateful to all data collectors for their commitments. Lastly, but not the least we thank rural women who have participated in this study for their priceless time.

## Author Contributions

**Conceptualization:** Wako Golicha Wako.

**Data curation:** Wako Golicha Wako, Belda Negesa Beyene.

**Formal analysis:** Wako Golicha Wako, Belda Negesa Beyene.

**Funding acquisition:** Wako Golicha Wako, Belda Negesa Beyene.

**Investigation:** Wako Golicha Wako, Belda Negesa Beyene.

**Methodology:** Wako Golicha Wako, Belda Negesa Beyene.

**Project administration:** Wako Golicha Wako, Belda Negesa Beyene.

**Resources:** Wako Golicha Wako, Belda Negesa Beyene.

**Software:** Wako Golicha Wako, Belda Negesa Beyene.

**Supervision:** Wako Golicha Wako, Belda Negesa Beyene.

**Validation:** Wako Golicha Wako, Belda Negesa Beyene, Zelalem Jabessa Wayessa, Aneteneh Fikrie, Elias Amaje.

**Visualization:** Wako Golicha Wako, Belda Negesa Beyene, Zelalem Jabessa Wayessa, Aneteneh Fikrie, Elias Amaje.

**Writing – original draft:** Wako Golicha Wako, Belda Negesa Beyene, Zelalem Jabessa Wayessa, Aneteneh Fikrie, Elias Amaje.

**Writing – review & editing:** Wako Golicha Wako, Belda Negesa Beyene, Zelalem Jabessa Wayessa, Aneteneh Fikrie, Elias Amaje.

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
