## [Decision Letter · Decision Letter 0]

3 Sep 2021

 PGPH-D-21-00293 Assessment of neonatal thermal cares: practices and beliefs among rural women in West Guji zone, south Ethiopia: a cross-sectional study PLOS Global Public Health

Dear Dr. Wako,

Thank you for submitting your manuscript to PLOS Global Public Health. After careful consideration, we feel that it has merit but does not fully meet PLOS Global Public Health’s publication criteria as it currently stands. Therefore, we invite you to submit a revised version of the manuscript that addresses the points raised during the review process. 

We look forward to receiving your revised manuscript.

Kind regards,

Nicola Hawley

Academic Editor

Journal Requirements:

Additional Editor Comments (if provided):

Please see the PLOS Data Policy and consider whether the data underlying this work can be made publicly available. See specifically the FAQ on why single authors are not recommended as the point of contact for work submitted.Please consider adding to the conclusions section of your abstract the possible public health implications of your research findings.Please provide detail about the women who were not successfully interviewed - did they decline participation or were they not contacted? If they declined, were there any key differences between the sample who chose to participate and those who declined?  

Reviewers' comments:

Reviewer's Responses to Questions

**Comments to the Author**

1. Does this manuscript meet PLOS Global Public Health’s publication criteria? Is the manuscript technically sound, and do the data support the conclusions? The manuscript must describe methodologically and ethically rigorous research with conclusions that are appropriately drawn based on the data presented.

Reviewer #1: Yes

Reviewer #2: Partly

2. Has the statistical analysis been performed appropriately and rigorously?

Reviewer #1: Yes

Reviewer #2: I don't know

3. Have the authors made all data underlying the findings in their manuscript fully available (please refer to the Data Availability Statement at the start of the manuscript PDF file)?

Reviewer #1: Yes

Reviewer #2: No

4. Is the manuscript presented in an intelligible fashion and written in standard English?

Reviewer #1: Yes

Reviewer #2: No

5. Review Comments to the Author

Reviewer #1: This paper is interesting to read and publishable. It has some grammatical errors, that can be worked on. I have also added some comments that should be reviewed and updated before publishing. This will be new information for scientists to learn from

Reviewer #2: Thanks for the opportunity to revise this paper and congratulations to the authors for submitting it. This paper has interesting findings on newborn thermal care practices that could be potentially used to improve newborn care and prevent newborn mortality. It also sheds light on the local practices of a population barely present in the international literature. However, some further work is needed to make it suitable for publication. Here are my recommendations

General: The main issue I see is that the paper was supposed to “assessed practices about neonatal thermal cares (early drying and wrapping, head covering, skin to skin care, newborn bath time and early breastfeeding initiation) and related beliefs among rural women in West Guji Zone, South Ethiopia”. I do not think that the “related beliefs” have been deeply explained except for a few quotes referring to cold prevention. An exception to this is the paragraph on lack of skin-to-skin care. There is no clear understanding of why participants think that wrapping and drying the baby is important or why it has to be done after the first bath. These types of reasons are mentioned in the survey results (Tables 3, 4, and 5) but the main reason to have a qualitative arm of a project like this is to explore these reasons and the customs/beliefs attached to it. I also recommend that the paper is proofread by a native English speaker.

Abstract: I think you meant “quantitative” when you wrote: “The qualitative data were cleaned, coded, and analyzed by SPSS version 20”.

Introduction:

1. In the end, the authors say “Few studies have assessed neonatal thermal care practices in rural Ethiopia”. Could you cite some of these? How is your study different than those?

2. Either here or in the discussion, could you mention if there are institutional (governmental or non-governmental) programs or policies aimed at improving newborn thermal care?

3. Explain the “Harmful cultural norms and traditional practices that affect neonatal thermal care practices at home exist in different contexts of Africa[14]”. What are the known challenges in Ethiopia or the region regarding neonatal thermal care?

Methods:

1. Sampling procedures: what are the different climate areas? Why are they important to determine a sampling process? Do they correspond to different ethnicities? Different language? Since we are talking about thermal care practices, can the differences in climate between these areas affect how they take care of their newborns’ temperature after being born?

2. Data collection instruments and procedures: how were the FGDs participants recruited? Did any of the participants receive compensation for their time? Were the FGDs conducted in Afaan Oromo? Is that the native language of all the FGDs participants? Is the principal investigator fluent in that language? To what language were the transcripts translated? How long were the FGDs? Where did they take place?

3. Quality assurance: “All errors in transcription and translation those were found were corrected.” What type of errors are you referring to? In what context were transcripts revised? During coding?

Results:

1. Based on the results in Tables 3, 4, and 5, and the TGDs guide (File 2), the authors seem to have some interesting data on why women have the newborn thermal care practices that they have. Even if that is part of the goals of the paper, it is not analyzed carefully. Authors have done a great job describing when newborn thermal care practices happen but little is done to explain the reasons either practical (e.g., the baby has to wait to be dried because there is no one to do it while the placenta is being delivered) or cultural behind it. I think the authors should have a section on that where they combine both the qualitative and quantitative data about why people wait until the first bath to apply thermal care practices.

2. I recommend more consistency in how results are presented. For example, in the first paragraph of Women’s Practice of Neonatal Thermal Care, they discuss immediate drying and include both quantitative and qualitative data. However, in the following paragraphs when they discuss other neonatal thermal care practices, they don’t include any qualitative data.

3. Based on my past two comments, I recommend that the results section is restructured. A first subsection could include what participants report doing regarding neonatal thermal care practices in both the survey and the TGDs. This would look like their first paragraph of the Women’s Practice of Neonatal Thermal Care, meaning a combination of rates and quotes about their drying, wrapping, and bathing practices as well as when/if immediate breastfeeding and skin-to-skin care take place. In a separate subsection, I recommend they include the reasons why women perform their neonatal thermal care practices the way they do. To explain this I suggest authors use both the reasons that came up in the survey and those from the TGDs. I also encourage the authors to note the differences between the participants’ responses in the survey and the TGDs. For example, keeping the baby from being cold seems to be the main reason for giving the baby an early bath in the qualitative data, but this does not appear in Table 4 (unless it is included in the “To improve health and strength of newborn” category. Even if that is the case, only 5.5% of participants reported that and it seems much more prevalent in the quotes). This could be further discussed in the Discussion section.

4. Are there any more reasons behind covering the newborn’s head than preventing cold?

5. Head covering is done after the first bath? Before? During?

6. How common among participants were the quotes you provided in the paper? It would be helpful to know if they are representative of most participants and what unique/negative cases came up.

7. If skin-to-skin contact is not a common practice, how do mothers hold their babies? Did you ask that?

8. What happened with the baby between birth and the first bath? Are they kept naked? are they hold by someone or placed somewhere? Is that surface clean? Are they wrapped and then unwrapped for the first bath?

9. Why does the first bath take place after the delivery of the placenta?

10. Unclear: “Women strongly believe that newborns should be bathed as soon as possible after delivery provided that placental delivery is not delayed. Other recommended newborn thermal care like immediate drying and wrapping are usually done after the first newborn’s bath opposite to the standard recommendation. However, newborn’s bath is rarely done before delivery of placenta”.

11. Were there any differences in the participants’ qualitative and quantitative responses based on age, climate area, economic status, religion, and educational level? Whether if there were differences or not, I think it is worth mentioning in the Results sections and even discussing later.

12. Did you ask about where did the water for bathing the baby came from (e.g., boiled water? Potable water? River water?)? I am afraid that could compromise the health of the newborn too.

13. “However, only 32(8.2%) of women dried-off their babies of birth fluids before

delivery of placenta”. Women did or their birth attendants? Who is in charge of wrapping, drying, and bathing the newborn?

Discussion:

1. In the first paragraph of the discussion it is worth mentioning that even though drying and wrapping are newborn thermal care practices, they are conducted after the first bath. In case that order compromises the efficiency of those thermal care practices in preventing hypothermia, it should be noted and supported with scientific literature.

2. You mention some of that at the end of the second paragraph: “they believe better thermal protection would be achieved if drying and wrapping are preceded by bath”. However, I suggest you also put it in the first paragraph because you are summarizing your findings there. I also recommend a discussion about the risks of drying and wrapping the baby after an early bath, if any.

3. Regarding paragraph 3, are there any other studies about head covering that you could cite and compare to your results as you did in paragraphs 2 and 4?

4. The paragraph on early bathing (#5) should be placed after paragraph 2 to continue the discussion about early bathing.

5. In the skin-to-skin care paragraph (#6) I suggest you discuss the reasons found in other studies to explain the lack of skin-to-skin care.

6. Limitations should include that the results cannot be generalized to the population.

7. I suggest you include a paragraph with recommendations to improve the neonatal thermal care practices in the region studied. Especially for delaying the early first bath and incorporating skin-to-skin care.

Conclusion: There is no mention of your findings regarding immediate breastfeeding.

6. PLOS authors have the option to publish the peer review history of their article (what does this mean?). If published, this will include your full peer review and any attached files.

**Do you want your identity to be public for this peer review?** For information about this choice, including consent withdrawal, please see our Privacy Policy.

Reviewer #1: **Yes: **Geoffrey Babughirana

Reviewer #2: No

---

## [Decision Letter · Decision Letter 1]

22 Nov 2021

PGPH-D-21-00293R1

Assessment of neonatal thermal cares: practices and beliefs among rural women in West Guji zone, south Ethiopia: a cross-sectional study

Dear Dr. Wako,

Thank you for submitting your manuscript to PLOS Global Public Health. As you will see the reviewer's noted a marked improvement in the manuscript but suggested some additional, minor edits before consideration for publication.

We look forward to receiving your revised manuscript.

Kind regards,

Nicola Hawley

Academic Editor

Journal Requirements:

Additional Editor Comments (if provided):

Reviewers' comments:

Reviewer's Responses to Questions

**Comments to the Author**

1. If the authors have adequately addressed your comments raised in a previous round of review and you feel that this manuscript is now acceptable for publication, you may indicate that here to bypass the “Comments to the Author” section, enter your conflict of interest statement in the “Confidential to Editor” section, and submit your "Accept" recommendation.

Reviewer #2: (No Response)

2. Does this manuscript meet PLOS Global Public Health’s publication criteria? Is the manuscript technically sound, and do the data support the conclusions? The manuscript must describe methodologically and ethically rigorous research with conclusions that are appropriately drawn based on the data presented.

Reviewer #2: Partly

3. Has the statistical analysis been performed appropriately and rigorously?

Reviewer #2: I don't know

4. Have the authors made all data underlying the findings in their manuscript fully available (please refer to the Data Availability Statement at the start of the manuscript PDF file)?

Reviewer #2: Yes

5. Is the manuscript presented in an intelligible fashion and written in standard English?

Reviewer #2: Yes

6. Review Comments to the Author

Reviewer #2: (No Response)

7. PLOS authors have the option to publish the peer review history of their article (what does this mean?). If published, this will include your full peer review and any attached files.

**Do you want your identity to be public for this peer review?** For information about this choice, including consent withdrawal, please see our Privacy Policy.

Reviewer #2: **Yes: **Irene Del Mastro N.

---

## [Decision Letter · Decision Letter 2]

30 Mar 2022

PGPH-D-21-00293R2

Assessment of neonatal thermal cares: practices and beliefs among rural women in West Guji zone, south Ethiopia: a cross-sectional study

Dear Dr. Wako,

Thank you for submitting your manuscript to PLOS Global Public Health. After careful consideration, we feel that it has merit but does not fully meet PLOS Global Public Health’s publication criteria as it currently stands. Therefore, we invite you to submit a revised version of the manuscript that addresses the points raised during the review process.

Your paper has been reviewed by the same Reviewers and - as you will see in their comments - there remain concerns about your presentation of the results section of the paper. I agree with the issues that the reviewer raises but believe with some careful editing the paper can be improved enough for publication. Please follow the wonderful advice of the reviewer (without taking their instructions word for word) and attempt to revise that section. It may be worth considering the services of someone who can look at the revision before resubmission, since copyediting services are not provided post-acceptance.

We look forward to receiving your revised manuscript.

Kind regards,

Nicola L. Hawley

Academic Editor

Journal Requirements:

Additional Editor Comments (if provided):

Reviewers' comments:

Reviewer's Responses to Questions

**Comments to the Author**

1. If the authors have adequately addressed your comments raised in a previous round of review and you feel that this manuscript is now acceptable for publication, you may indicate that here to bypass the “Comments to the Author” section, enter your conflict of interest statement in the “Confidential to Editor” section, and submit your "Accept" recommendation.

Reviewer #2: (No Response)

2. Does this manuscript meet PLOS Global Public Health’s publication criteria? Is the manuscript technically sound, and do the data support the conclusions? The manuscript must describe methodologically and ethically rigorous research with conclusions that are appropriately drawn based on the data presented.

Reviewer #2: Partly

3. Has the statistical analysis been performed appropriately and rigorously?

Reviewer #2: I don't know

4. Have the authors made all data underlying the findings in their manuscript fully available (please refer to the Data Availability Statement at the start of the manuscript PDF file)?

Reviewer #2: No

5. Is the manuscript presented in an intelligible fashion and written in standard English?

Reviewer #2: No

6. Review Comments to the Author

Reviewer #2: See attachment

7. PLOS authors have the option to publish the peer review history of their article (what does this mean?). If published, this will include your full peer review and any attached files.

**Do you want your identity to be public for this peer review?** For information about this choice, including consent withdrawal, please see our Privacy Policy.

Reviewer #2: No

---

## [Editor Report · Decision Letter 3]

11 May 2022

Assessment of Neonatal Thermal Cares: Practices and Beliefs among Rural Women in West Guji Zone, South Ethiopia: A Cross-sectional Study

PGPH-D-21-00293R3

Dear Mr Wako,

We are pleased to inform you that your manuscript 'Assessment of Neonatal Thermal Cares: Practices and Beliefs among Rural Women in West Guji Zone, South Ethiopia: A Cross-sectional Study' has been provisionally accepted for publication in PLOS Global Public Health.

Best regards,

Nicola L. Hawley

Academic Editor